# Disarm The Bacteria: What Temperate Phages Can Do

Shiyue Zhou [1,2,†], Zhengjie Liu [2,†], Jiaoyang Song [2] and Yibao Chen [2,*]

1   College of Veterinary Medicine, Huazhong Agricultural University, Wuhan 430070, China
2   Institute of Animal Science and Veterinary Medicine, Shandong Academy of Agricultural Sciences, Jinan 250100, China
*   Correspondence: yibaochen2012@126.com; Tel./Fax: +86-531-6665-5093
†   These authors contributed equally to this work.

**Abstract:** In the field of phage applications and clinical treatment, virulent phages have been in the spotlight whereas temperate phages received, relatively speaking, less attention. The fact that temperate phages often carry virulent or drug-resistant genes is a constant concern and drawback in temperate phage applications. However, temperate phages also play a role in bacterial regulation. This review elucidates the biological properties of temperate phages based on their life cycle and introduces the latest work on temperate phage applications, such as on host virulence reduction, biofilm degradation, genetic engineering and phage display. The versatile use of temperate phages coupled with their inherent properties, such as economy, ready accessibility, wide variety and host specificity, make temperate phages a solid candidate in tackling bacterial infections.

**Keywords:** temperate phages; host virulence reduction; biofilm degradation; genetic engineering and modification

## 1. Introduction

A bacteriophage (hereafter phage) is the virus that parasitizes bacteria, and is recognized as for its ability to lyse its host. As the most prevalent and widely distributed group of viruses on earth, phages are estimated to be around $10^{31}$ in the biosphere [1]. Shortly after being discovered in the 1910s, phages drew attention on account of their therapeutic potential to treat infectious diseases [2]. Moreover, successful phage treatment had been performed on several infections such as those caused by *Shigella dysenteriae*, *Salmonella*, and *Escherichia coli* [3]. Regardless, due to the rapid rising of antibiotics, further development on phage therapy was interrupted [4]. In recent years, with the growing concern about antibiotic resistance and immediate needs for a more reliable substitute [5], research on phages as well as phage treatment is burgeoning again [6–10]. Antibiotics such as beta-Lactam have been confirmed to promote interbacterial gene transfer and, thus, potentially increase bacterial virulence [11]. Due to its merits, such as high specificity and accessibility, phage therapy emerges as a novel antibacterial counterweight under the inevitable trend of antibiotic misuse [6,12–14]. Compared to antibiotics, phages persist as biological entities and play a significant role in mediating and regulating the bacterial community, physiology and evolution which, furthermore, affects the ecological system [15]. For instance, phages' contribution to carbon cycling and bacteria diversity is of great value in the ecosystem [16]. Cell lysis induced by phages has a major impact on dissolved organic carbon turnover and nutrient cycling in food web processes and biogeochemical cycles [17]. Through gene transfer, phage infection casts a vast influence on the diversity of prokaryotic species [18]. Meanwhile, the constant contact between bacteria and phage closely affects the dynamic state of human intestinal microflora [19,20]. Thus, in-depth research and understanding should be carried out before phage therapy can be fully credited and broadly implemented.

Taxonomically, phages are categorized primarily according to their morphology and genome, and more refined and comprehensive classification is under way [21]. Based on

their lifecycle, phages are differentiated into virulent phages and temperate phages. In the first case phage reproduction always leads to host death [22], which is not necessarily the case for the second. The unique reproductive manner of temperate phages involves integration into the host genome and replication along with the host [23]. The integration renders a bacterium and temperate phage as lysogen and prophage, respectively [24,25]. In this review, we mainly pivot our attention to temperate phages. On account of their distinct way of living, a vast variety of interactions and mechanisms await exploration and employment. However, phage therapy often brings to mind the use of virulent phages (also known as lytic phages), referring to their ability to kill bacteria "quick and clean" [24]. Compared to virulent phages, temperate phages are in lack of sophisticated exploration and utilization in further clinical treatment. To fully understand and better employ the advantages of temperate phages, a thorough recognition of phage–bacteria interaction and the successful experiments based on these insights up to the present is needed.

## 2. Temperate Phage

Temperate phages are detected in a large proportion of bacteria [22]. When integrated to the host genome, the prophage genome can account for up to 20% of the bacteria genome [26]. A total of 46% of bacteria are estimated to be lysogens [27,28]. Among marine viruses, temperate phages are reckoned to be omnipresent in the *Vibrionaceae* family [29], members of which are often naturally endemic to warm marine and estuarine waters. There are abundant temperate phages that have lysogenized the most destructive fish pathogens, such as *Flavobacterium psychrophilum*, which currently causes considerable economic losses in salmonid aquaculture [30]. Around 50% of bacteria harbors at least one prophage [27], whereas the majority of phages only parasitize on one bacterium and retain high host specificity [6].

To date, phage lambda, which infects *E. coli*, is perhaps the most thoroughly studied and widely applied temperate phage [31]. The lambda DNA is double stranded, with a genome of about 50 Kb [32]. Genetically, phage lambda possesses "moron" genes with differential function, location, size and GC contents to the rest of phage genome [33]. Although these moron genes do not usually participate in the essential functioning of the phage lifecycle, their potential for providing a selective advantage is indubitable [25].

Phage lambda could reproduce many generations in a lysogenic cycle. The lytic-driving genes persist in prophage, yet repressed, which is the key lysogenic maintaining force [34]. Under certain stressors, such as antibiotics, this lysogenic status will be induced and switched into the lytic, which consequently leads to host lysis [35]. Prophage stability is attributed to the domestication of prophage elements within the host genome; however, there is also a case demonstrating that the lytic cycle could be restarted through certain DNA recombinations [36]. Once the phage lambda turns on its lytic cycle switch, it can produce around one hundred new virions from each bacteria lysis [37]. The transformation between two lifecycles provides temperate phages with more possibility of surviving [35].

Of note, the interactions between temperate phages and the host genome are complicated. Prophage-induced lysis can impose beneficial effects on the bacterial population and, thus, is preferred in the interest of the overall situation [38]. Most of the bacterial virulence genes are carried by temperate phages [39]. Prophage behaviors can give rise to or increase host bacteria virulence [40]. Nevertheless, there is also proven evidence of eliminated or decreased virulence as a result of temperate phage involvement [41,42]. For instance, a previous study showed that the integration site of PHB09 is specifically located within a pilin gene of *Bordetella bronchiseptica* and decreased the virulence of parental strain *B. bronchiseptica* Bb01 in mice [42]. vB_SauS_JS02 is a *Siphoviridae* temperate phage infecting *S. aureus* and shows stronger host inhibition activity than antibiotic ceftazidime [43].

## 3. Temperate Phage Life Cycle

Using lambda phage as a role model, we can obtain basic insights of the life cycle of temperate phages and the host–phage interrelationship. Temperate phage development

that may occur through the whole life cycle could be categorized into five major phases. (1) Diffusion: the phage gets through the biofilm to approach the host bacteria. (2) Absorption and injection: the phage binds with receptor proteins on the surface of the bacteria and injects the phage genome. (3) Integration and replication: the phage genome inserts into the bacteria genome and becomes a prophage, or persists independently as plasmid. (4) Induction and packaging: the prophage becomes activated and enters the lytic cycle. (5) Lysis of host bacterium: the progeny phage is released from the bacteria [16,44–46].

There is expansive diversity in each step to help the phage adapt to multiple conditions [47]. Upon each step, a choice has to be made for the phage (or prophage) to proceed precisely at the genetic level [35]. A number of factors determine the phage's reproduction circuits. These factors include the genome size of the host bacteria, population growth status and other bacterial pathogens, temperature, etc [27]. Crucially, the decision has to be made quickly and accurately, without any compromised or mixed states of cycles [35].

Here, we introduce the mechanism and interaction in the order of the temperate phage life cycle. Understanding the close relationship between the phage and host in a life-cycle order will help us to build a more holistic and intuitive perception towards phage antibacterial activity (Figure 1).

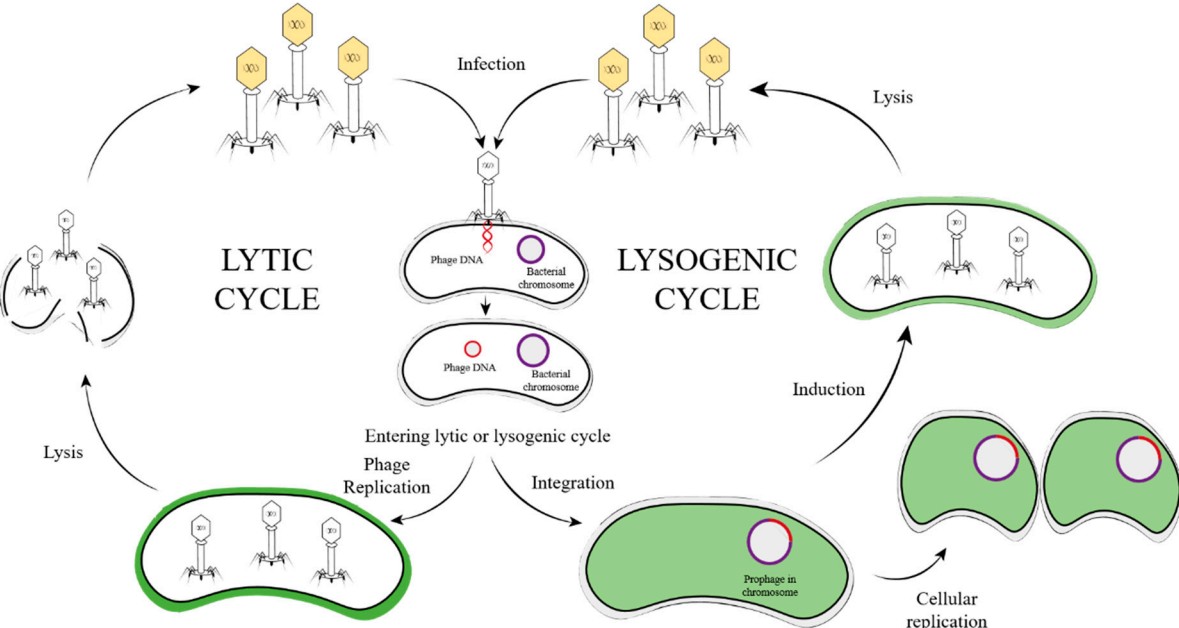

**Figure 1.** Life cycle of temperate phage. Temperate phage ejects its genome into host bacteria during infection and persists as either prophage or separate plasmid. For lysogenic life, temperate phage replicates along with host bacterium, until prophage induction is triggered and leads to host lysis.

### 3.1. Step One—Diffusion

In this initial phase, phages diffuse and penetrate through the biofilm to locate and target bacteria. Composed of polymeric substances and bacteria-secreted enzymes and proteins, the biofilm matrix wraps around the microbial community and serves as a powerful physical barrier against not only phages but also immune system and antimicrobial agents [48,49]. Bacterial biofilm formation and toxin production are indicators of bacterial virulence [50,51]. A plethora of chronic and difficult-to-treat infections are associated with biofilm formation [52]. Therefore, penetration through the biofilm is vital for a successful phage infection.

There is growing evidence supporting the theory that phages can promote biofilm formation [53,54]. However, a great part of the phages is reported to be able to encode and produce biofilm degrading enzymes, referred to as depolymerase [55], such as capsular polysaccharides, exopolysaccharides, and lipopolysaccharides [56]. Found mostly as part

of phage tail fiber or tail spike proteins, these depolymerases function as a weapon to depolymerize bacterial capsules and facilitate phage absorption [57].

### 3.2. Step Two—Absorption and Injection

When initiating infection, phages first adsorb to specific receptors on the surface of bacteria [58]. Bacteria can obtain phage resistance by deleting or inactivating phage-specific receptors [59]. The binding of phages to their receptors exerts selective pressure on bacteria, which alters the expression of the receptors, thereby preventing phage infection [60]. If phage receptors are associated with virulence factors or antibiotic resistance mechanisms in the target bacteria, such fitness trade-offs might reduce the virulence or antibiotic resistance of the pathogenic bacteria [61].

However, most of the temperate phages are limited in their receptor binding proteins (RBPs) and are obligated to specific hosts [58,62]. Evolutionarily selected pressure as well as artificial genetic engineering by swapping genes can increase the number of RBPs [63]. For a group of phages with different RBPs, certain combinations of those phages can achieve the goal of expanding receptor range [64–68].

### 3.3. Step Three—Integration and Replication

Many phages encode an integrase that integrates phage DNA into the host chromosome [69]. Phages can exist as plasmids outside of the chromosome as well [70,71]. The highly regulated integration and excision process ensures the efficient and accurate switch between the lytic and lysogenic pathway [72]. Not only does prophage influence the virulence of bacteria, it also regulates the host gene expression. The excision of prophage in *Listeria monocytogenes* is key in escaping cell phagosomes [73].

Lysogenic conversion is the process of prophage gene expressing as a part of host genome and is reciprocal for the prophage and host [74]. Prophages can affect bacterial infectivity, toxin secretion, virulence regulation, surface modification, immune stimulation and evasion, and microbiome competition [75,76]. The insertion of phage genomes in the bacterial genome can disrupt biofilm formation-related genes, leading to a reduction in biofilm formation [76].

To make sure of their chance of survival, phages develop a number of defense mechanisms to eliminate unwanted host-sharing. The known mechanisms are divided into three groups: blocking genome injection, expressing repressor protein, and binding inhibition [77]. In addition, phage-inducible chromosomal islands (PICIs) can also provide impressive protection to resident phages against other intruding phages and mobile elements with the aid of helper phages [78].

With regard to phage–phage interaction within one same host, intricate coordination can be achieved upon triggering SOS responses by two phages sharing one host. In *L. monocytogenes* strain 10403S, the two coexisting prophages regulate simultaneous induction and lytic activity under SOS conditions. Moreover, the host can also benefit from the cooperation of its habitants. To maintain harmonious coexistence of two prophages, AriS is discovered as a conserved phage protein and is demonstrated to be capable of avoiding SOS response and phage induction by inhibiting RecA [79].

### 3.4. Step Four—Induction and Packaging

Prophages can be induced into the lytic cycle under a series of stressors, such as antibiotics and UV rays [80]. Some of them trigger selective induction, and only certain prophages can be induced [81]. Pyocyanin produced by *Pseudomonas aeruginosa* exhibits a selective induction to phage phiMBL3 [82]. In addition, the bacterial SOS response induces prophage in a non-selective manner. The SOS response is a survival strategy when facing stressors that may endanger the host and damage DNA [83]. Separated from the SOS response, prophage induction can also be triggered by chemicals such as acyl-homoserine lactones under high bacteria density condition [84].

Normally, the progeny phage is packaged, released, and keeps infecting the next host [85]. However, this process of packaging is also accident-prone. Transduction, also known as phage-mediated gene transfer, occurs in three different mechanisms: specialized transduction, generalized transduction, and lateral transduction. Both specific transduction and general transduction are the result of phage mispackaging [86]. Generalized transduction happens when a phage accidentally packages random bacteria DNA, which can be "uploaded" in the next eligible host [87]. In comparison, specialized transduction is usually favored by host bacteria on account of the possible acquisition of phage DNA that may contribute to host fitness and virulence [88]. Lateral transduction happens when DNA packaging begins with delayed excision of intact and functional prophages. It also means that the prophage genomes remain integral to the host genome while they are replicated, which eventually leads to the presence of multiple copies of the phage genome in the host genome. Such transduction mechanism results in a higher frequency of host DNA transfer [89].

### 3.5. Step Five—Lysis of Host Bacteria

Single-stranded DNA phages cause hydrolysis of the host bacterial cell wall by synthesizing enzymes that interfere with host bacterial peptidoglycan synthesis, whereas double-stranded DNA phages, such as phage lambda, hydrolyze the host bacterial peptidoglycan structure by lysin or endolysin, which are synthesized late in replication [90].

In a nutshell, the above steps illustrate the life cycle of temperate phages, using the lambda phage as an example. The interactions between temperate phages and their bacterial hosts are complex and intimate. While new mechanisms are being discovered, we should also look at how we can better exploit the intimate relationship between phage and bacteria to help in the fight against pathogenic bacteria.

## 4. Temperate Phages for Therapeutic Purposes

As we discussed in the temperate phage life cycle, the transduction and lysogenic conversion may cause undesirable outcome such as virulence promotion (Table 1). For instance, the production of temperate phage Pf can lead to a significant virulence increase in infections in its host, *Pseudomonas*, and may affect the entire lung ecosystem [91]. Mice infected with *Pseudomonas* strains that are deleted of Pf4 prophage survive significantly longer, which indicates that the presence of prophage Pf4 is a virulence contributor [92]. Furthermore, prophages without virulence genes may result in virulent effects on host bacteria, and increased virulence can occur without virulence-related genes. Lysogenic MRSA strains (SA14+) exhibited improved virulence, stress tolerance, and biofilm-forming abilities when a temperate PHB21 containing no virulence gene was inserted [93]. A change in temperate phage status in host cells can also lead to unwanted consequences. For instance, the entire gut virome analysis supports that virome changes are associated with inflammatory bowel disease patients' guts. A transition from lysogenic to lytic replication in the gut may result in inflammatory bowel disease (IBD) [94]. In the gut of autism spectrum disorder patients after Microbiota Transfer Therapy treatment, an altered phage community coupled with increased bacteria community diversity is observed, suggesting the putative role of phages in gut dysbiosis [95].

**Table 1.** List of temperate phages with their favorable characteristics described.

| Phage | Host | Function | Description | Reference |
|---|---|---|---|---|
| PHB09 | *B. bronchiseptica* | Virulence shrinks under massive phage predation | PHB09 inserted and thus disrupted pilin protein gene, but the vaccine made of lysogenic *B. bronchiseptica* strain Bb01+ also showed effective protection of mice challenged with virulent *B. bronchiseptica*. | [42] |
| p2 | *K. pneumoniae* | Host prevents invasion by reducing virulence | The presence of a plasmid form of prophage can provide host bacterium with resistance to other foreign DNA at the cost of the host virulence. | [96] |
| ΔLCRA500 | *L.monocytogenes* | Host prevents invasion by reducing virulence | The temperate phage ΔLCRA500, which has been knocked out of the gp32, gp33 and integrase genes, has marked lytic ability and a specific *Listeria* serotype 4b host range. | [97] |
| PHB22a, PHB25a, PHB38a, and PHB40a | Methicillin-Resistant *S. aureus S-18* | Temperate phage cocktails enhanced with ions | The antibacterial effect of this recipe is determined by the biofilm removal efficiency, where added ions proved higher bacterial CFU reduction ability. Moreover, using *G. mellonella* larvae as animal model against MRSA S-18 infection, the survival rate resulting from ions–phages therapy is 10% higher. | [65] |
| SA13m | *S.aureus* | Converted into stable lytic phage | A virulent mutant SA13m obtained through random deletion of temperate phage SA13 exhibits active lytic activity and no sign of lysogenicity. Application of SA13m in sterilized milk showed that *S. aureus* was reduced to non-detectable levels, suggesting that SA13m can efficiently control the growth of *S. aureus* in food. | [75] |
| AP3 | *B.cenocepacia* | Combined with antibiotics | Temperate *Burkholderia* phage AP3 combined with antibiotics demonstrates increased bactericidal effects in in vivo experiments with moth larvae. | [98] |
| M13 | *C. trachomatis* | Temperate phage display | Compared to *C. trachomatis* infection alone, engineered phages stably express RGD motifs and *C. trachomatis* peptides and significantly reduce *C. trachomatis* infection in HeLa and primary cervical cells. | [99] |
| 933W | *E. coli* | Modification of phage genes to inhibit toxin production | The phage demonstrated superior toxin inhibition in both in vivo and in vitro infections. In the foodborne pathogen EHEC, the λ prophage 933W both produces Stx2 and inhibits phage overlap infection of other λ phages. | [41] |
| Eλ | *E. coli* | Gene-modified phage with CRISPR-Cas3 system | A genetically engineered λ phage exhibits enhanced killing ability and host specificity when incorporated with CRISPR-Cas3 system and knockdown of the lytic gene cro. This engineered phage specifically and effectively eliminates enterohemorrhagic *E. coli* infection and validated the superior performance over wild-type phages through in vitro and in vivo experiments. In addition, there is no evidence in this study showing that EHEC developed resistance to engineered lambda phage. | [100] |
| HK97 | *E. coli* | Combined with antibiotics | In vitro bacterial eradication is observed after coadministration of *E. coli* temperate phage HK97 and antibiotic ciprofloxacin. This synergy works in line with the depletion of lysogens which ciprofloxacin specially targets. | [101] |

**Table 1.** *Cont.*

| Phage | Host | Function | Description | Reference |
|---|---|---|---|---|
| λ | *E. coli* | Combined with antibiotics | The restoration of antibiotic sensitivity to two antibiotics, streptomycin and nalidixic acid, can be realized by the introduction of specific genes rpsL and gyrA, respectively, in the process of temperate phage lysogenization. | [102] |
| λ | *E. coli* | Phage vaccine | Recombined with targeted DNA, phage λ can carry the particulate DNA into human system and become protected from degradation, making sure the antigen presenting cells can recognize and capture them. | [103] |
| λ | *E. coli* | Phage vaccine | A vaccine made from temperate phage λ using phage display technique showed significant efficiency in eliciting anti-PCV2 immune response after the first vaccination without adjuvant. | [104] |
| M13 | *E. coli* | Phage vaccine | Using temperate phage M13 surface display, the diverse clone of tumor-associated antigens in prostate cancer is achieved and makes it a desirable candidate for vaccine development in prostate cancer. | [105] |
| Filamentous phage | *E. coli* | Phage vaccine | The filamentous phage inoculation induced both humoral and cellular immune response against HSV-1 in BALB/c mice. | [106] |
| λ | *E. coli* | Lambda PLP | Phage-like particles (PLPs) are derived from phage lambda, and robust internalization of Trz PLPs resulted in increased intracellular Trz concentrations, prolonged cell growth inhibition and regulation of cellular programs associated with HER2 signaling, proliferation, metabolism and protein synthesis compared to Trz treatment. | [107] |
| λ | *E. coli* | Reverse antibiotic sensitivity | Using lysogenic conversion, a sensitivity cassette is brought into the bacteria genome and unwanted recombination is managed to be avoided. | [102] |
| DMS3 | *P.aeruginosa* | Encode proteins that block QS system | *P. aeruginosa* phage DMS3 can protect bacteria from attack by other phages by inhibiting bacterial quorum sensing. DMS3 encodes a QS anti-activator protein aqs1 that is expressed immediately after phage infection. aqs1 inhibits the activity of LasR, a major regulator of quorum sensing, and restrains twitching motility and superinfection. | [108] |
| LKA1 | *P.aeruginosa* | Lyase production to eradicate biofilm | A temperate phage of *Pseudomonas* has been proved to be able to produce a lyase, LKA1gp49, to degrade LPS. LKA1gp49 lyase efficiently reduces *P. aeruginosa* virulence in the in vivo *G. mellonella* infection model, and sensitizes bacterial cells to the lytic activity of serum complement. | [109] |
| Ef11 | *E. faecalis* | Converted into stable lytic phage | By deletion of putative lysogeny gene module and replacement of putative cro promoter from the recombinant phage genome with a 50 nisin-inducible promoter, the temperate phage is rendered virulent and with expanded host range. | [110] |
| 3A2 | *R. parkeri* | Gene insertion led to attenuated phenotype | The *R. parkeri* mutant strain is genetically modified by inserting a transposon into the gene encoding the phage integrase in the bacterial genome. Such a mutant exhibited significantly reduced virulence, significantly smaller phage plaques and improved histopathological alterations in intravenously infected mice compared to the parental wild type. | [111] |

**Table 1.** *Cont.*

| Phage | Host | Function | Description | Reference |
|---|---|---|---|---|
| *Gardnerella* phage | *G. vaginalis* | Engineered endolysins | A genetically modified endolysin PM-477 produced by *Gardnerella* phage exhibits the ability to completely disrupt bacterial biofilms of *G. ardnerella vaginalis* and has no effect on beneficial *Lactobacillus* or other species of vaginal bacteria. | [112] |
| ZoeJ and BPs | *M. abscessus* | Converted into stable lytic phage | Two temperate phages are transformed into lytic phages and made into a three-phage cocktail along with one lytic phage. The cocktail is administered to a cystic fibrosis patient and recovering signs are observed after six months' treatment. | [113] |
| BP96115 | *Salmonella* | Virulence shrinks under massive phage invasion | As opposed to the streptomycin treatment, pre-treatment of mice with temperate phage safeguarded a stable and more diverse gut ecosystem and protected the intestinal system of mice against the pathogen challenge. | [114] |

Nevertheless, temperate phages also have their advantages that cannot be ignored. A wide variety of temperate phages are found in nature and can be easily induced in the laboratory [115]. Temperate phages are easier to obtain than lytic phages and, as biotechnology matures, temperate phages can be modified and assigned specific properties [116]. Here, we discuss some successful applications of temperate phages to reduce host bacterial virulence. There are a handful of methods that can be applied to achieve the goal, including biofilm degradation, phage cocktails with expanded host ranges, genetic engineering, and phage display (Table 1).

### 4.1. Host Virulence Reduction

Temperate phages have been widely demonstrated to promote bacterial virulence, which is the most important factor in causing infections [117]. However, there is no lack of evidence indicating that phages can reduce bacterial virulence, intrinsically or after proper modification. For example, reducing virulence to gain phage resistance, or reducing virulence to increase host fitness (Figure 2A).

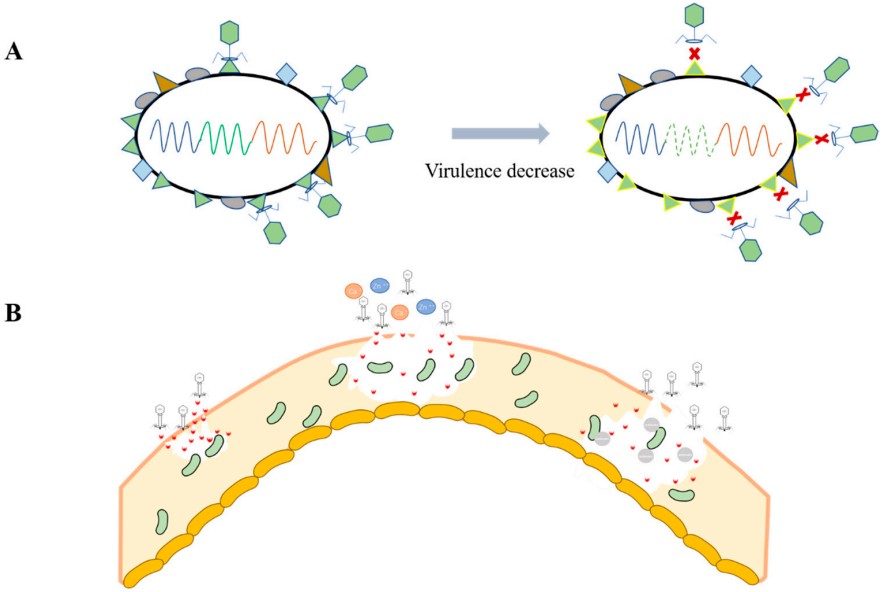

**Figure 2. Different angles of temperate applications.** (**A**)Trade-off between virulence and phage resistance. Temperate phages can reduce bacteria virulence through a virulence–phage resistance

trade-off mechanism. Facing massive predation, bacteria mutate their surface protein gene and avoid temperate phage binding. At the same time, the virulence decreases as the cost of the mutation. (**B**) Biofilm degradation. Certain biofilm degrading enzymes produced by temperate phages can accelerate the penetration towards target bacteria. With the addition of metal ions and antibiotics, the bactericidal effect is enhanced.

**Hosts prevent invasion by reducing virulence**. The presence of prophage in the form of plasmid can provide host bacteria with resistance to other foreign DNA at the cost of host virulence. p2 is proposed to be an intact plasmid prophage in *Klebsiella pneumoniae*. Mutant Kp1604Δp2 exhibits an increase in host virulence which is determined by mouse infection models. The mutant p2 minus strain leads to 100% mortality compared to the 70% mortality of the p2 carrying strain, indicating that the presence of p2 decreases the virulence of its host [96].

In *L. monocytogenes*, there is a classic trade-off situation. The cell wall of *Listeria* and its associated proteins are responsible for most of the interactions with the mammalian host [118]. The temperate phage ΔLCRA500, which has been knocked out of the gp32, gp33 and integrase genes, has marked lytic ability and a specific *Listeria* serotype 4b host range. In order to prevent phage adsorption, *L. monocytogenes* 4b underwent a mutation associated with phosphoribonate glycosylation, resulting in the loss of galactose from the phosphoribonate molecule. This loss of galactose not only prevented phage adsorption, but also led to a reduction in bacterial virulence [97].

**Virulence shrinks under massive phage predation**: A phage mixture targeting different extracellular structures such as receptors causes a huge impact on the bacterial fitness, virulence, and pathogenicity of *P. aeruginosa*. Secretory virulence factors, such as elastase, pyocyanin, and pyoverdine, significantly facilitate the *P. aeruginosa* colonization of new niches but are not directly related to the cell response to phage infection. The change in *P. aeruginosa* PAO1 biology is related to the number of phages that cause selection pressure on the population. The more phages appear in the environment, the deeper and more noticeable are the phenotypic changes involving a reduction of various virulence factors' production levels [119].

To further investigate temperate phages' potential to benefit clinical treatment, temperate phage PHB09's interaction with its host *B. bronchiseptica* Bb01 has been elucidated in detail. Isolated from sewage water, temperate phage PHB09 is reported to attenuate host virulence by lysogenization. Not only does the temperate phage reduce the virulence of its host both in vivo and in vitro, most likely by inserting and thus disrupting the pilin protein gene, but the vaccine made of lysogenic *B. bronchiseptica* strain Bb01+ also showed effective protection of mice challenged with virulent *B. bronchiseptica*. Moreover, in the sight of possible risks rising from prophage induction and phage releasing, neither antibiotic resistance genes nor reversion of bacteria virulence are observed. All induced bacteria are lysed eventually. Thus, the successful virulence attenuation of prophage PHB09 proposes a promising frontier of temperate phages being developed as vaccines [42].

Conducted by Bao's team, an alleviating effect of temperate phage pre-treatment is observed on intestinal dysbiosis and inflammation in challenged mice. As opposed to the streptomycin treatment, the pre-treatment of mice with temperate phages safeguarded a stable and more diverse gut ecosystem and protected the intestinal system of mice against the pathogen challenge [120].

*4.2. Biofilm Degradation*

Biofilms can inhibit drug penetration, and thereby significantly reduce the killing efficiency of antimicrobials [114]. Biofilms can also help bacteria to adhere better to the site of infection, causing chronic infection. Thus, in terms of reducing biofilm formation, temperate phages can produce enzymes that degrade biofilms [109], and this degradation can be enhanced by the addition of certain ions [65]. From this point of view, temperate

phages have a promising application in the treatment of pathogenic bacterial infections (Figure 2B).

**Temperate phage cocktails enhanced with ions**. Biofilm can provide a sanctuary for bacteria being hunted by antibiotics. The thick matrix of biofilm formed by bacteria is a special shield against antibiotics because of the reduced drug penetration and the accessibility [121]. A phage cocktail consisting of four temperate phages of the *Siphovirdae* family, administered with metal ions $Ca^{2+}$ and $Zn^{2+}$, shows the enhanced bactericidal impact both in vitro and in vivo compared to the phage cocktail alone. In this experiment, the reason ions such as $Ca^{2+}$ and $Zn^{2+}$ can offer an advantage to the phage cocktail is likely because of the promoted fluidity and stability of cocktail phages in the biofilm. The possible gene transfer through these temperate phages is avoided thanks to the lack of virulence gene in the studied four phages. In addition, by decreasing host virulence, prophage can also be a good helper in outcompeting other bacteria. The antibacterial effect of this recipe is determined by the biofilm removal efficiency, where added ions proved a higher bacterial CFU reduction ability. Moreover, using *Galleria mellonella* larvae as animal model against MRSA S-18 infection, the survival rate resulting from ions–phages therapy is 10% higher than the phage cocktail alone [65].

**Temperate phage encoded enzyme to eradicate biofilm**. A temperate phage of *Pseudomonas* has been proved to be able to produce a lyase, LKA1gp49, to degrade LPS. LKA1gp49 is a lyase that degrades the O5-serotype specific polysaccharide. This enzyme degrades LPS molecules embedded in the cell envelope and disperses the biofilm matrix, resulting in an increased diffusion rate for small molecules. LKA1gp49 lyase efficiently reduces *P. aeruginosa* virulence in the in vivo *G. mellonella* infection model and sensitizes bacterial cells to the lytic activity of the serum complement. LKA1gp49 could also be a potential additive for antimicrobials, as it does not interrupt the efficacy of ciprofloxacin and gentamicin [109].

**Combined with antibiotics:** Temperate phage can provide enhancement efficacy in killing bacteria with antibiotics [122,123]. The outcome is dependent on the type of phage, type of antibiotics and their respective concentration. Therefore, the results alter dramatically even with one small change of the elements [124]. For instance, *Burkholderia cenocepacia* is one of the most important opportunistic pathogens in causing high mortality rates in cystic fibrosis (hereafter CF) patients [125]. CF is a chronic genetic disease, caused by a loss in the gene to keep osmotic balance [126,127]. The thick mucus formed as a result of osmotic imbalance blocks airways in the respiratory system and makes it difficult for the antibiotic to penetrate and reach the infecting bacteria [128]. With universal awareness of its antibiotic resistance, finding a suitable antimicrobial therapy substitute is paramount.

Temperate *Burkholderia* phage AP3 combined with antibiotics demonstrates increased bactericidal effects in in vivo experiments with moth larvae. This finding could be considered as a potent lead against bacterial strains belonging to *B. cenocepacia* IIIA lineage, which are commonly isolated from CF patients [98]. Attention should be paid to antibiotic resistance change when practicing temperate phage therapy in CF patients. *P. aeruginosa* is the cause of a typically challenging infection endocarditis. Increased resistance to antibiotics and a broadened host range of *P. aeruginosa* is observed, along with disease progression [129].

For temperate phages, the prophage induction through the SOS response and resensitization to antibiotics are the two main synergy mechanisms [130]. In the work of Amany M Al-Anany, in vitro bacterial eradication is observed after the coadministration of *E. coli* temperate phage HK97 and antibiotic ciprofloxacin. This synergy works in line with the depletion of lysogens which ciprofloxacin specially targets [101]. The restoration of antibiotic sensitivity to two antibiotics, streptomycin and nalidixic acid, can be realized by the introduction of specific genes, rpsL and gyrA, respectively, in the process of temperate phage lysogenization [102].

## 5. Temperate Phage Gene Engineering and Display

In addition to functioning as a killing machine, a temperate phage can be a powerful and stable vector, for instance, carrying vaccines into bacteria and acting as a medium for protein expression. The engineering of a phage integration site or the inhibition of the toxins' gene expression can also take a toll on host virulence. Temperate phages are a robust platform for genetic engineering and modification [131].

**Phage vaccine**: Temperate phage lambda's potential for delivering a DNA vaccine has been exploited and substantiated [132]. It is economical, and has excellent stability, easy production and, most importantly, no concern of antibiotic resistance [133]. With all these appealing advantages, temperate phage is a promising candidate for vaccines. Recombined with targeted DNA, temperate phage λ can carry particulate DNA into the human system and provide protection from degradation, making sure the antigen presenting cells can recognize and capture them. A phage vaccine has only been administered by sub-cutaneous and intramuscular injections, but an oral form of delivery is also possible dependent on its stability in water [103]. A vaccine made from temperate phage λ using the phage display technique showed significant efficiency in eliciting an anti-PCV2 immune response after the first vaccination without adjuvant [104]. Using temperate phage M13 surface display, the diverse cloning of tumor-associated antigens in prostate cancer is achieved and makes it a desirable candidate for vaccine development in prostate cancer [105]. Vaccines consisting of filamentous phage are also considered a viable alternative. The filamentous phage inoculation induced both humoral and cellular immune responses against HSV-1 in BALB/c mice [106].

**Lambda PLP**: Phage-like particles (PLPs) derived from phage lambda have physico-chemical properties compatible with drug standards, and in vitro particle tracking and cellular targeting is achieved by displaying fluorescein-5-carboximide (F5M) and trastuzumab (Trz), respectively. Phage-derived nanodrugs are modular systems that can be easily adapted to combined approaches, including imaging, biomarker targeting and the intracellular delivery of therapeutics. A 'designer nanoparticle' system that can be rapidly engineered in a tunable and unambiguous manner, trz-PLP binds to oncogenically active human epidermal growth factor receptor 2 (HER2) and is internalized by HER2 overexpressing subtypes of breast cancer cells, but not by breast cancers lacking HER2 amplified breast cancers. The robust internalization of Trz PLPs resulted in increased intracellular Trz concentrations, prolonged cell growth inhibition and the regulation of cellular programs associated with HER2 signaling, proliferation, metabolism and protein synthesis compared to Trz treatment. The robustness and flexibility of lambda PLP provides a platform that adapts to a wide range of utility and customized features [107].

**Engineered endolysins**: Bacteriophage-derived endolysins are cell wall hydrolases which could hydrolyze the peptidoglycan layer from inside and outside bacterial pathogens [134]. A genetically modified endolysin PM-477 produced by *Gardnerella* phage exhibits the ability to completely disrupt bacterial biofilms of *Gardnerella vaginalis*. *G. vaginalis* is a common vaginal bacterium, but can cause bacterial vaginosis under abnormal growth. This engineered endolysin PM-477 has a strong specificity and efficiency against *Gardnerella* strains, and has no effect on beneficial *Lactobacillus* or other species of vaginal bacteria [112].

**Temperate phage display**: The temperate phage M13 phage has a wide range of applications in biomedical materials, and is used for different therapeutic applications [135]. This is due to its unique biological characteristics: safety, ready modification and the ability to form nanofiber shapes and self-assemble into nanofiber matrices. Bhattarai's team has engineered and edited an M13 phage, a phage carrying two functional peptides; the integrin binding peptide (RGD), and a polymorphic membrane protein D (PmpD) fragment from *Chlamydia trachomatis*, a globally prevalent human pathogen for which there is no effective approved vaccine [136]. Compared to *C. trachomatis* infection alone, engineered phages stably express RGD motifs and *C. trachomatis* peptides and significantly reduce *C. trachomatis* infection in HeLa and primary cervical cells [99].

**Gene insertion led to attenuated phenotype**: The *Rickettsia parkeri* mutant strain is genetically modified by inserting a transposon into the gene encoding the phage integrase in the bacterial genome. Such a mutant exhibits significantly reduced virulence, significantly smaller phage plaques and improved histopathological alterations in intravenously infected mice compared to the parental wild type. Furthermore, single-dose intradermal immunization of this mutant strain provided mice with complete protection against the lethal *R. parkeri* rickettsioses in mice. Such a live attenuated rickettsial mutant strain could be used as a novel potential vaccine candidate for the treatment of spotted fever rickettsial disease [111].

**Modification of phage genes to inhibit toxin production**: Produced by some *E. coli*, Shiga toxin (Stx) is causative of gastrointestinal diseases and hemolytic uremic syndrome with high incidence and lethality [137]. Shiga-producing *E. coli* is one of the four pathogens among the mostly benign intestinal commensal *E. coli* strains [138]. Notably, the virulence factor related to Shiga toxin production is introduced by two lambda-like prophages, which are one of the main genetic elements in causing virulence.

To curb toxin release, the temperate phage λ was genetically engineered to express a deterrent that neutralizes Stx production in *E. coli*, and the genetic mosaicism of the λ phage was exploited to create a hybrid phage capable of overcoming the phage resistance mechanism. The phage demonstrated superior toxin inhibition in both in vivo and in vitro infections. In the foodborne pathogen EHEC, the λ prophage 933W both produces Stx2 and inhibits phage overlap infection of other λ phages [41].

**Gene-modified phage with CRISPR-Cas3 system:** CRISPR (Clustered Regularly Interspaced Short Palindromic Repeats)-Cas (CRISPR associated) system is a defense system in bacteria, possessed by around 40% of bacteria [139]. To exclude foreign DNA from infecting bacteria, bacteria initiate the CRISPR-Cas system by recognizing, "memorizing", foreign DNA, and make targeted cleavage upon reoccurring infection attempt [140]. Temperate phage infection can cause maladaptive immunopathological effects on its host and lead to self-targeting of the CRISPR-Cas system, which inhibits the growth of host bacteria [141]. In type 1 Crispr-Cas system, the cascade complexes show a remarkable fast speed in scanning DNA sequences and providing protection [142].

A genetically engineered lambda phage exhibits enhanced killing ability and host specificity when incorporated with the CRISPR-Cas3 system and knockdown of the lytic gene cro. This engineered phage specifically and effectively eliminates enterohemorrhagic *E. coli* infection and validates the superior performance over wild-type phages through in vitro and in vivo experiments. In addition, there is no evidence in this study showing that EHEC developed resistance to an engineered lambda phage [100].

**Encode proteins that block the QS system:** The Quorum Sensing (QS) system is a communication system amongst bacteria to modulate community behaviors, a regulatory system that controls the expression of virulence factors and secreted public goods [143]. These circuits enable bacteria to measure the density of their neighbors via receptors. The population-sensing receptors then activate a signaling cascade that leads to global transcriptional changes [144]. Moreover, temperate phages can benefit from QS signals. In *vibriophage* VP882, through spying on the host-produced anti inducers during QS process, VP882 is capable of manipulating its own cycle switch [145].

*P. aeruginosa* phage DMS3 can protect bacteria from the attack of other phages by inhibiting bacterial quorum sensing. DMS3 encodes a QS anti-activator protein aqs1 that is expressed immediately after phage infection. aqs1 inhibits the activity of LasR, a major QS regulator, and restrains twitching motility and superinfection. Although there is a 100-fold increase in the number of cells killed by DMS3aqs1 infection compared to wild-type DMS3 infection, no more phages were produced. This suggests a role for anti-phage mechanisms. Aqs 1 offers a counterstrategy through which phages might simultaneously silence multiple antiphage defenses [108].

**Converted into the stable lytic phage**: A virulent mutant SA13m obtained through the random deletion of temperate phage SA13 exhibits active lytic activity and no sign

of lysogenicity. The application of SA13m in sterilized milk showed that *S. aureus* was reduced a non-detectable levels, suggesting that SA13m can efficiently control the growth of *S. aureus* in food [75]. Two temperate phages are transformed into lytic phages and made into a three-phage cocktail along with one lytic phage. The cocktail is administered to a cystic fibrosis patient and recovering signs are observed after six months' treatment [113]. An *Enterococcus faecalis* temperate phage is converted to a lytic phage for therapeutical purposes. By the deletion of the putative lysogeny gene module and replacement of the putative cro promoter from the recombinant phage genome with a 50 nisin-inducible promoter, the temperate phage is rendered virulent and with expanded host range [110].

## 6. Conclusions

Temperate phages, because of their natural biological properties, play an integral and indispensable role in the war between phage and bacterium. Temperate phages have contributed a variety of new genetic resources to the bacterial gene pool. Found in half of the bacteria, temperate phages have more accessibility than virulent phages [146]. With proper intervention, or purposeful selection such as genetic engineering, temperate phages are a powerful tool to combat bacterial infections by delivering vaccines and degrading biofilm. Therefore, the study of complex but highly resilient interrelationships between phages and bacteria is of great importance. In this review, we organized previous and recent studies as well as demonstrated empirical research. Mounted cases have highlighted the fact that temperate phage recognition and utilization is heading towards the right direction.

However, there is also room for improvement, for instance in the underutilization of the induction of hidden lysogenic phages. The actual employment of induced temperate phages can be more technically demanding, considering the need to remove integrase and integrate related genes. Targeted embedding of phages to disrupt virulence genes is a promising direction for research, of course making sure that the phages do not carry any virulence genes or integrases.

In summary, this paper focuses on the life cycle of temperate phages and the interactions they have with their host bacteria. Most importantly, this paper illustrates an array of temperate phage applications we could employ in order to combat bacterial infection and benefit clinical treatment. Many experiments have demonstrated the great efficacy and usefulness of temperate phages in the treatment of bacterial diseases, but more in-depth studies are yet to be discovered.

**Author Contributions:** Software, J.S.; formal analysis, Y.C.; writing—original draft preparation, S.Z., Z.L. and J.S.; writing—review and editing, S.Z.; project administration, Y.C.; funding acquisition, Y.C. All authors have read and agreed to the published version of the manuscript.

**Funding:** This study was funded by grants from the Innovation Capability Improvement Project for Science and Technology SMEs in Shandong Province (2022TSGC2384) and the Natural Science Youth Foundation of Shandong Province (ZR2022QC028), and Linyi City Agricultural and Animal Husbandry Waste Recycling and Public Health Improvement Project (CXGC2022A27).

**Institutional Review Board Statement:** Not applicable.

**Informed Consent Statement:** Not applicable.

**Data Availability Statement:** Not applicable.

**Conflicts of Interest:** The authors declare no conflict to interest.

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
