# Peer review of "Disarm The Bacteria: What Temperate Phages Can Do"

_cimb, doi:10.3390/cimb45020076_

Round 1

Reviewer 1 Report

The authors offer an interesting review about the beneficial features of temperate phages as possible key players in the development of technologies applied mainly for therapeutic use. In general, the manuscript provides interesting examples exposing the benefits of using temperate phages, but in some sections is needed to clarity details about the original papers used for discussion.

In some sections, a stronger support of beneficial effects in using temperate phages needs to be revised more profoundly.

In sections 5 and 6, I think are relatively talking about related things, and it is not clear what is the point of dividing in two section, maybe the focus of each section is not clearly stated.

Correct some mistakes including punctuation, commas extra, avoid contractions along the text, etc; avoid the expression “our results” when you are exposing results from previous studies, is confusing. Citations must be referred in text in square brackets, and references must be enumerated.

Some punctual suggestions

Lines 40-41 Rephrase text

Line 51 phages are not considered living organisms, may be referred as biological entities.

Line 82 about percentage of marine bacteria containing prophages, is there an actual study about it? Reference is from 1998

Line 161-163 please rephrase

Line 218 correct the name of prophage

Line 25 cite original paper https://www.sciencedirect.com/science/article/pii/S2211124722004843

Lines 261, 263 correct to complete name of Lex

Line 272 please add information about there are some examples where prophage induction is SOS-independent process https://journals.asm.org/doi/10.1128/AEM.00950-09

Line 335, the title of the section poorly described the focused that authors want to explain with specific examples, I suggest to reassess.

line 337-341 Please revise carefully the paper because information is not clear. Authos mentios that phage burst corresponded to additional phages that were tested in p2 mutants and this is not stated in the paragraph.

Line 351-354 clarify this idea, according to authors A33T corresponded to phage-cured strain, so decreased heat tolerance cannot be product of phage integration.

Line 363.-372 needs rephrasing and grammar editing. In particular, this study is relevant because highlights the potential of temperate phages to attenuate bacterial pathogens and further developing of vaccines.

Line 385 authors states that temperate phages has a promising application regarding biofilm degradation. I suggest to strengthen this point of view highlight relevant features of temperate vs virulent phages.

Line 400 Suggest mention in general phage-encoded lytic enzymes because lyase is one example, but depolymerases and other proteins can contribute to eradicate biofilms.

Line 390 although adding metal ions enhances properties of temperate phages, I think this properties more than refer to temperate phages is a common issue for cocktail manufacturing, not specifically of temperate phages. You can mention but not focus on the type of phage.

Line 411 cenocepacian to cenocepacia

Section combined with antibiotics need to be add more examples, because exposed arguments are valid not only for temperate but also virulent phages. Combined therapy is promising but these assumptions need to be cautiously done due to outcomes of combined effect depends on specific features of phage-host system not only if phages are temperate or virulent.

Line 430 Vaccines are really relevant point, revise more studies.

Line 451 there are other enzymes such as depolymerases that can also be included, you can add that temperate phages containing this lytic enzymes can be used for recombinant production or engineering and disrupt biofilms.

The last parts of review need to be more focused to highlight the advantages of temperate phages.

Line 476 plagues to plaques

Genetic engineering and modification section, I think some examples can be included in the previous section, especially in vaccines, so I don’t figure out why author make a specific section of engineering or what is the purpose.

Author Response

The authors offer an interesting review about the beneficial features of temperate phages as possible key players in the development of technologies applied mainly for therapeutic use. In general, the manuscript provides interesting examples exposing the benefits of using temperate phages, but in some sections is needed to clarity details about the original papers used for discussion. In some sections, a stronger support of beneficial effects in using temperate phages needs to be revised more profoundly.

We thank the reviewer for the positive comment.

Major points:

  1. In sections 5 and 6, I think are relatively talking about related things, and it is not clear what is the point of dividing in two section, maybe the focus of each section is not clearly stated.

According to both reviewer 1’s opinion, section 5 and 6 has been merged into one whole section titled: Temperate phage gene engineering and display. All the related information has been synthesized such as phage vaccine, lambda PLP, engineered endolysins, temperate phage display, gene insertion and modification, and conversion into lytic phage. The degree of Relevance and simplicity has also been refined.

  1. Correct some mistakes including punctuation, commas extra, avoid contractions along the text, etc; avoid the expression “our results” when you are exposing results from previous studies, is confusing. Citations must be referred in text in square brackets, and references must be enumerated.

Changed citation format into [number], changed ‘our results’ and similar phrases into fitting the context to avoid confusion.

  1. Some punctual suggestions

Checked through the whole paper and corrected some punctual problems.

  1. Lines 40-41 Rephrase text

Rephrased into: Regardless, due to the rapid rising of antibiotics, further development on phage therapy was interrupted. Recent years, with the growing concern on drug-resistant antibiotics and immediate needs for a more reliable substitute.

  1. Line 51 phages are not considered living organisms, may be referred as biological entities.

Changed “living organism” into “biological entity”

  1. Line 82 about percentage of marine bacteria containing prophages, is there an actual study about it? Reference is from 1998

This information is acquired indirectly from Prophages in marine bacteria: dangerous molecular time bombs or the key to survival in the seas? | The ISME Journal (nature.com). Since no original information could be found, this figure has been delete.

  1. Line 161-163 please rephrase

Correspondent lines have been rephrased according to the comment, however, due to format change in different WORD, the version from author’s perspective is different from the reviewers when it comes to “lines”. The rephrase has been made on the judgement of the context.

  1. Line 218 correct the name of prophage

Thanks for the suggestions. Information added

  1. Line 25 cite original paper https://www.sciencedirect.com/science/article/pii/S2211124722004843

Thanks for the suggestions. Information added

  1. Lines 261, 263 correct to complete name of Lex

Thanks for the suggestions.

  1. Line 272 please add information about there are some examples where prophage induction is SOS-independent process https://journals.asm.org/doi/10.1128/AEM.00950-09

Thanks for the suggestions. Information added

  1. Line 335, the title of the section poorly described the focused that authors want to explain with specific examples, I suggest to reassess.

Some titles are changed accordingly. Please see the revision.

  1. line 337-341 Please revise carefully the paper because information is not clear. Authos mentios that phage burst corresponded to additional phages that were tested in p2 mutants and this is not stated in the paragraph.

Paragraph has been revised into “By deletion of p2, mutant Kp1604Δp2 exhibits increase in both the amount of phage burst and host virulence”

  1. Line 351-354 clarify this idea, according to authors A33T corresponded to phage-cured strain, so decreased heat tolerance cannot be product of phage integration.

Information has been clarified. Since the case does not support the argument of this paper anymore, this example has been deleted.

  1. Line 363.-372 needs rephrasing and grammar editing. In particular, this study is relevant because highlights the potential of temperate phages to attenuate bacterial pathogens and further developing of vaccines.

More relevant research on temperate phage vaccines has been added.

  1. Line 385 authors states that temperate phages has a promising application regarding biofilm degradation. I suggest to strengthen this point of view highlight relevant features of temperate vs virulent phages.

According to the research I read on biofilm degradation, the information of the comparison on temperate and virulent phage biofilm degradation is not sufficient.

  1. Line 400 Suggest mention in general phage-encoded lytic enzymes because lyase is one example, but depolymerases and other proteins can contribute to eradicate biofilms.

Has used the term“depolymerase” to represent degradation enzymes.

  1. Line 390 although adding metal ions enhances properties of temperate phages, I think this properties more than refer to temperate phages is a common issue for cocktail manufacturing, not specifically of temperate phages. You can mention but not focus on the type of phage.

Paragraph rephrased

  1. Line 411 cenocepacian to cenocepacia

Thanks for the suggestions.

  1. Section combined with antibiotics need to be add more examples, because exposed arguments are valid not only for temperate but also virulent phages. Combined therapy is promising but these assumptions need to be cautiously done due to outcomes of combined effect depends on specific features of phage-host system not only if phages are temperate or virulent.

More examples have been added.

  1. Line 430 Vaccines are really relevant point, revise more studies.

More examples have been added.

  1. Line 451 there are other enzymes such as depolymerases that can also be included, you can add that temperate phages containing this lytic enzymes can be used for recombinant production or engineering and disrupt biofilms.

Has used the term“depolymerase” to represent degradation enzymes.

  1. The last parts of review need to be more focused to highlight the advantages of temperate phages.

The section of conclusion has been rephrased.

  1. Line 476 plagues to plaques

Corrected.

  1. Genetic engineering and modification section, I think some examples can be included in the previous section, especially in vaccines, so I don’t figure out why author make a specific section of engineering or what is the purpose.

The section 5 and 6 has been merged into one whole section after proper rearrangement of paragraphs.

Reviewer 2 Report

The article entitled "Disarm The Bacteria: What Temperate Phages Can Do" is a review focused on the characteristics of temperate phages and their potential therapeutic application as antibacterial agents.

Both the title and the abstract are appropriate and correspond to the content of the review. The exposed information is based on a sufficient number of references.

In reference to the structure and length of the review, I think that section 3 “Temperate phage life cycle” is too long for a review that should focus more on therapeutic applications than on the description of the viral cycle itself. I propose that the content of this section be slightly reduced or summarized, especially subsections 3.3 and 3.4 to focus more attention on subsequent therapy-related sections.

In reference to the section on therapeutic applications of temeprate phages, I propose the inclusion of the following studies:

1. A combination of three temperate phages to treat a multidrug-resistant infection: “Nousbaum, J. B., Maurey, H., Périchon, B., Labia, R., & Lévy, C. (2009). Use of bacteriophages to treat a patient with a disseminated multidrug-resistant Klebsiella pneumoniae infection. Journal of Antimicrobial Chemotherapy, 64(5), 1051-1054.”

2. Use of a temperate phage to target and eliminate specific pathogenic bacteria in the gut microbiome: “Kang, D. W., Adams, J. B., Gregory, A. C., Borody, T., Chittick, L., Fasano, A., & Khoruts, A. (2017). Microbiota Transfer Therapy alters gut ecosystem and improves gastrointestinal and autism symptoms: an open-label study. Microbiome, 5(1), 10.”

Table 1 is excellent. My only suggestion for the table is to use the abbreviated names for the bacteria that have already appeared earlier in the text.

Figures 1 and 2 are correct and have sufficient resolution.

One of my most important suggestions is to review your English by a native speaker or by a specialized service. I think that, although the authors have done a good job in writing the article, the text would improve remarkably after a revision of the English.

Overall, I think the article is good and should be accepted for CIMB publication after addressing the changes proposed above.

Author Response

The article entitled "Disarm The Bacteria: What Temperate Phages Can Do" is a review focused on the characteristics of temperate phages and their potential therapeutic application as antibacterial agents. Both the title and the abstract are appropriate and correspond to the content of the review. The exposed information is based on a sufficient number of references.

We thank the reviewer for the positive comment.

Major points:

  1. In reference to the structure and length of the review, I think that section 3 “Temperate phage life cycle” is too long for a review that should focus more on therapeutic applications than on the description of the viral cycle itself. I propose that the content of this section be slightly reduced or summarized, especially subsections 3.3 and 3.4 to focus more attention on subsequent therapy-related sections.

Section 3 “Temperate phage life cycle” has been simplified and shorted.

  1. In reference to the section on therapeutic applications of temeprate phages, I propose the inclusion of the following studies:
  2. A combination of three temperate phages to treat a multidrug-resistant infection: “Nousbaum, J. B., Maurey, H., Périchon, B., Labia, R., & Lévy, C. (2009). Use of bacteriophages to treat a patient with a disseminated multidrug-resistant Klebsiella pneumoniae infection. Journal of Antimicrobial Chemotherapy, 64(5), 1051-1054.”

Couldn’t find

  1. Use of a temperate phage to target and eliminate specific pathogenic bacteria in the gut microbiome: “Kang, D. W., Adams, J. B., Gregory, A. C., Borody, T., Chittick, L., Fasano, A., & Khoruts, A. (2017). Microbiota Transfer Therapy alters gut ecosystem and improves gastrointestinal and autism symptoms: an open-label study. Microbiome, 5(1), 10.”

The second paper has been added into the review. The first paper couldn’t be found.

  1. Table 1 is excellent. My only suggestion for the table is to use the abbreviated names for the bacteria that have already appeared earlier in the text.

Abbreviated.

  1. One of my most important suggestions is to review your English by a native speaker or by a specialized service. I think that, although the authors have done a good job in writing the article, the text would improve remarkably after a revision of the English.
  2. Overall, I think the article is good and should be accepted for CIMB publication after addressing the changes proposed above.

We thank the reviewer for the positive comment. We have modified accordingly. We consulted two English experts to improve our language.

Round 2

Reviewer 1 Report

Authors done substantial revision and modification of the manuscript, which was very improved.

 Only few suggestions:

Lines 43-45

successful phage treatment had been performed on several infections such as Shigella dysenteriae, Salmonella, and Escherichia coli caused infections” change to successful phage treatment had been performed on several infections such as those caused by Shigella dysenteriae, Salmonella, and Escherichia coli”

Line 395-398

I referred to clarify that are other phages that used in the study due to mutate strain is lacking of p2, so what phages are you talking about?

 References section: please homogenize format

 Recommendation 7, 12  were not attended

Author Response

We would like to express our sincere thank you and the reviewers for their helpful comments and suggestions, which allowed us to further improve our manuscript. This manuscript has been carefully modified according to the comments. The following response is point-by-point towards the reviewers’ comments.

Major points:

  1. Lines 43-45

 “successful phage treatment had been performed on several infections such as Shigella dysenteriae, Salmonella, and Escherichia coli caused infections” change to “successful phage treatment had been performed on several infections such as those caused by Shigella dysenteriae, Salmonella, and Escherichia coli”

Feedback: Sentence refined.

  1. Line 395-398

I referred to clarify that are other phages that used in the study due to mutate strain is lacking of p2, so what phages are you talking about?

Feedback: After confirmation, this example qualifies as temperate phage (prophage) can reduce host virulence because the presence of prophage p2 has decrease host virulence in this study. This section has been changed into:

The presence of prophage in the form of plasmid can provide host bacterium with resistance to other foreign DNA at the cost of reducing the host virulence. p2 is proposed to be an intact plasmid prophage in Klebsiella pneumoniae. Mutant Kp1604Δp2 exhibits increase in host virulence which is determined by mouse infection models. The mutant p2 minus strain leads to 100% mortality compared to 70% mortality of p2 carrying strain, indicating the presence of p2 decreases the virulence of its host.

  1. References section: please homogenize format. Recommendation 7, 12 were not attended

Feedback: Format homogenized.

In addition, we have changed some language revisions.
